# A Rare Case of Methemoglobinemia after Ifosfamide Infusion in a 3-Year-Old Patient Treated for T-ALL

**DOI:** 10.3390/ijms25073789

**Published:** 2024-03-28

**Authors:** Maria Suprunowicz, Katarzyna Marcinkiewicz, Elżbieta Leszczyńska, Anna Krętowska-Grunwald, Marcin Płonowski, Mariola Tałałaj, Łucja Dakowicz, Maryna Krawczuk-Rybak, Małgorzata Sawicka-Żukowska

**Affiliations:** 1Department of Pediatric Oncology and Hematology, Medical University of Bialystok, 15-274 Bialystok, Poland; 2Department of Anesthesiology and Intensive Care for Children and Adolescents with Postoperative and Pain Treatment Unit, Medical University of Bialystok, 15-274 Bialystok, Poland

**Keywords:** acute lymphoblastic leukemia, ALL, methemoglobinemia, ifosfamide, methylene blue, exchange transfusion

## Abstract

Methemoglobinemia is a potentially life-threatening, rare condition in which the oxygen-carrying capacity of hemoglobin is diminished. We present the case of a 3-year-old boy treated for T-cell acute lymphoblastic leukemia (T-ALL) who developed methemoglobinemia (MetHb 57.1%) as a side effect of ifosfamide administration. Due to his critical condition, the patient was transferred to the intensive care unit (ICU). The therapy included methylene blue administration, an exchange transfusion, catecholamine infusion, and steroids. Improving the general condition allowed for continuing chemotherapy without ifosfamide and completion of the HR2 block. Vigilance for methemoglobinemia as a very rare side effect should be widespread when using ifosfamide in the treatment protocols.

## 1. Introduction

Methemoglobinemia is a rare medical condition caused by the change of heme iron from reduced ferrous (Fe2+) to oxidized ferric (Fe3+) [1]. This results in the loss of the ability to attach and carry oxygen by hemoglobin, which leads to hypoxia. Methemoglobinemia is also called “functional anemia” due to the correct concentration of hemoglobin and the simultaneous loss of its function [2].

The methemoglobin (MetHb) level in the blood of a healthy person persists at around a maximum of 1% [3], while its concentration above 20% results in the occurrence of symptoms such as dyspnea, chest pain, headache, dizziness, tachycardia, seizures, central nervous depression, and even death. Therefore, it requires therapeutic actions [4,5]. To keep MetHb concentrations low, cytochrome b5 reductase (CYB5R) reduces MetHb to hemoglobin (Hb) with nicotinamide adenine dinucleotide (NAD) as a cofactor. Another pathway involves the enzyme nicotinamide adenine dinucleotide phosphate (NADPH)-MetHb reductase [6]. NADPH, a cofactor, requires glucose 6-phosphate dehydrogenase (G6PD). CYB5R is responsible for the removal of 95% to 99% of endogenously produced MetHb, while (NADPH)-MetHb reductase is responsible for the removal of only 5% (Figure 1) [1,3]. Yet, methylene blue (MB) can increase the pharmacological potency of this enzyme and accelerate MetHb metabolism [1].

The causes of methemoglobinemia can be congenital or acquired, although the acquired ones seem to occur more frequently [2]. Patients with the hereditary form of methemoglobinemia can tolerate high levels of MetHb without pronounced symptoms [3]. Inherited dysfunctions are due to either autosomal recessive variants in the *CYB5R3* gene or rarely the *CYB5A* gene, or else to autosomal dominant variants in the globin genes called M group variants of Hb with mutations (*HBA1*, *HBA2*), beta-globin (*HBB*), or gamma-globin (*HBG1*, *HBG2*) [5,7]. Methemoglobinemia can occur in patients with glucose-6-phosphate dehydrogenase (*G6PD*) deficiency treated with oxidative stressors. An X-linked gene mutation causes the enzymopathy, which disrupts the function of the sole enzyme in erythrocytes that produces NADPH, the metabolic intermediate essential for the maintenance of a high ratio of reduced to oxidized glutathione that protects erythrocytes from reactive oxygen species-mediated damage [8]. In comparison, acquired causes of methemoglobinemia are due to exposure to chemicals that cause a change in the degree of oxidation of hemoglobin, which include toxins and drugs like direct oxidizing agents such as benzocaine, lidocaine, and prilocaine, indirect oxidation, for example, nitrates, or metabolic activators like aniline and dapsone [9]. Other medications used during oncology treatment and possibly causing methemoglobinemia are metoclopramide [10], flutamide [11], silver nitrate [12], sulfonamides, including trimethoprim-sulfamethoxazole [13], cyclophosphamide [14], and rasburicase [15]. When treating tumor lysis syndrome with rasburicase in leukemia patients, the chance of acquiring methemoglobinopathy at the beginning of the illness is significant [16]. Medications that cause methemoglobinemia can also lead to hemolysis [5].

Methemoglobinemia can be diagnosed in patients with cyanosis not caused by cardiovascular disorders and presenting as increased MetHb, chocolate-colored blood, and normal PO2 values in arterial blood [17,18]. In patients with known methemoglobinemia, therapeutic management should be introduced: intravenous hydration, oxygen therapy, MB, vitamin C, and N-acetylcysteine, as well as blood transfusions, exchange transfusions, and hyperbaric therapy [5].

We report a case of a 3-year-old boy diagnosed with methemoglobinemia caused by ifosfamide during the T-cell acute lymphoblastic leukemia (T-ALL) treatment.

## 2. Case Description

A 3-year-old boy with T-ALL was admitted to the Department of Pediatric Oncology and Hematology, Bialystok, Poland, for the continuation of the chemotherapy block HR-2 (extended consolidation according to protocol AIEOP-BFM-2017 POLAND). The medical diagnosis of T-ALL was based on a bone marrow biopsy with the following cell phenotypes: CD5%—97%, CD7%—96%, CD8%—54%, CD2—94%, and cytCD3%—98%. The karyotype of the child was not performed. Due to poor prednisone response (PPR) and positive minimal residual disease (MRD) in TP1 (time point 1), he was classified in the HR group. During treatment preceding the episode of methemoglobinemia, no serious and unexpected side effects of treatment were observed; only myelosuppression, mucositis, and steroids-related mood disorders were observed.

The patient was in good general condition, with no indication of infection. During the morning rounds on the third day of the HR2 block of chemotherapy, the patient was in good condition. However, half an hour after the fourth dosage of ifosfamide (800 mg/m^2^/dose p.i. (1 h) every 12 h on days 2 to 4 (5 doses)—the patient’s body surface is 0.71 m^2^, he received 550 mg), unforeseen symptoms appeared. The child presented with severe pallor with cyanosis of the skin, blue lips, fine papular rash, sobbing, and nervousness with HR 120–190/min, RR 105/56 mmHg, and oxygen saturation 76–89% (attempts made at applying passive oxygen insufflation were unsuccessful due to the child’s anxiety). 

The medications used in the HR-2 block treatment, except ifosfamide, were dexamethasone p.o., vindesine i.v., methotrexate i.v., daunorubicin i.v., PEG-Asparaginase i.v., and ondansetron. None of the above medications have been linked to the possibility of methemoglobinemia. One drug that is well-known to cause methemoglobinemia is trimethoprim-sulfamethoxazole, used to prevent Pneumocystis jirovecii in patients treated for ALL [13]. However, our patient received prophylactic treatment in between blocks of chemotherapy, and no complaints were observed from any administration of this medication. The symptoms presented by our patient occurred immediately after the fourth dose of ifosfamide administration. The patient did not receive ifosfamide in any subsequent chemotherapy block, but further treatment, including cyclophosphamide infusions, was continued without side effects. 

Additionally, hydrocortisone, clemastine, metamizole, and furosemide were given. Imaging tests were performed with no signs of pulmonary embolism. Due to the child’s deteriorating health condition and saturation dropping to 70%, the patient was admitted to the intensive care unit (ICU) with the symptoms of severe respiratory failure. 

Upon admission to the ICU, the patient’s condition was critical, which required intubation and mechanical ventilation in PCV mode with PIP-22 cmH_2_O, FiO_2_ 1.0, PEEP 5.0, RR 30/min, Sat 70%, and HR 82/min. In the spectrophotometric method called co-oximetry (using a Roche cobas b 221 analyzer), hemoglobin concentrations were total hemoglobin (THb) 10.38 g/dL, oxyhemoglobin (O_2_Hb) 42.70%, deoxyhemoglobin (HHb) 0.00%, carboxyhemoglobin (COHb) 0.20%, and MetHb 57.10% (Table 1). The therapy included methylene blue 2 mg/kg in i.v. injection in a 5-min infusion (the patient’s body weight was 20 kg; he received 40 mg MB i.v. per dose). The drug was administered twice—before and after the exchange transfusion. Additionally, he received a catecholamine infusion (dobutamine, dopamine), an antibiotic, an antifungal, a diuretic, and steroids (hydrocortisone). At the time of the exchange transfusion, which was already after the administration of MB, the child’s blood was a dark chocolate color (Figure 2). 

MetHb concentrations in blood serum steadily decreased as a result of medication implementation, including MB administration and exchange transfusion with subsequent MB injection (MetHb 28%, MetHb 10.30%, respectively), with the level of MetHb being 0.70% after 24 h of primary evaluation. The child’s general condition progressively improved. After 2 days in the ICU, the boy was extubated, disconnected from the ventilator, and left with passive oxygen insufflation at 2 L/min. In stable overall condition, he was transferred to the Department of Oncology and Hematology for further treatment.

The last, fifth dosage of ifosfamide was not given; the rest of the treatment was maintained. The boy in good condition was discharged from the hospital after completing the chemotherapy HR-2 block. Further treatment included only cyclophosphamide, after which the boy had neither cyanosis nor other symptoms that would indicate methemoglobinemia. Before the episode of methemoglobinemia, the patient received three doses of 1 g/m^2^ CPM and five doses of 500 mg/m^2^ CPM without any symptoms of methemoglobinemia, an important alkylating drug used in the treatment of ALL. He continued treatment according to protocol with the use of cyclophosfamide without any symptoms. Although cyclophosphamide is also recognized as a substance that can cause methemoglobinemia [14], this complication was not observed in our patient. The child had received the drug in previous chemotherapy blocks and was not associated with any complaints suggesting the onset of methemoglobinemia. Continuation of cyclophosphamide treatment was dictated by the patient’s normal parameters, and on subsequent administration of the medication, continuously monitored saturation was at a normal level of 95–99%. There was no need to monitor methemoglobin levels. Currently, the child is undergoing oral maintenance treatment. The phenomenon of methemoglobinemia has not appeared again since the discontinuation of ifosfamide administration.

## 3. Discussion

Ifosfamide is an alkylating agent and cyclophosphamide analog, one of the chemotherapeutic agents used in many therapeutic protocols for pediatric oncological diseases, including the treatment of ALL, the most frequent childhood cancer in the pediatric population [19,20,21]. The most prevalent ifosfamide adverse effects include hemorrhagic cystitis, nephrotoxicity, myelosuppression, and neurotoxicity demonstrated as encephalopathy [4,21,22,23]. Methemoglobinemia is a very uncommon condition of ifosfamide treatment [24]. The mechanism by which the symptoms occurred after the administration of the third dose of ifosfamide in our patient remains unclear. In this case, the likely mechanism may have been the accumulation of the drug dose, as it was administered in short intervals every 12 h.

Cytochrome P450 (CYP) catalyzes the conversion of ifosfamide to 4-hydroxyifosfamide in the liver. The combination of 4-hydroxyifosfamide with its other form, aldoifosfamide, results in the final cytotoxic metabolites of ifosfamide: mustard and acrolein (Figure 3) [25,26]. Acrolein is known to cause hemorrhagic cystitis, while ifosfamide mustard is responsible for DNA alkylation, which causes cytotoxicity. Ifosfamide can also be deactivated, resulting in the release of a potentially neurotoxic chloroacetaldehyde. What is more, chloroacetaldehyde may decrease glutathione, which protects DNA against alkylation by ifosfamide mustard, at normal levels [25]. Ifosfamide mustard has a strong preference for erythrocytes; more of the active metabolite is carried in the red blood cells than in the plasma [27]. The role of the erythrocytes in the transport and release of active ifosfamide metabolites is unclear. It cannot be ruled out that red blood cells are capable of directly delivering factors to the tissue via fenestration [28].

Hadjiliadis et al. reported a case of a 58-year-old African American woman with stage IV metastatic leiomyosarcoma who developed minor dyspnea after the first dosage of ifosfamide administration, which disappeared following oxygen treatment. After 30 min of ifosfamide injection, the patient developed severe dyspnea, mental status changes, and disorientation, resulting in intubation and mechanical ventilation with MetHb 49.8%, which had increased to 56.6% in 30 min before therapy. In the treatment, MB was used, with a good response of MetHb level being 31.2% after 1 h to 1.4% after 24 h after treatment [24]. Hadjiliadis et al. hypothesize that methemoglobinemia was caused by concurrent phenobarbital administration, which enhanced ifosfamide metabolism and elevated ifosfamide toxic agents [24,25], which reduced glutathione stores in red blood cells (RBCs) [24]. Moreover, Valiev T.T. et al. described 14 cases, 5 of which were ALL pediatric patients who developed methemoglobinemia after receiving ifosfamide [29]. Methemoglobinemia emerged after the first dosage in 30% of patients but after subsequent doses in 70%. The symptoms were cyanosis, weakness, and a fall in saturation to 75–85%, with MetHb values ranging from 21% to 38.9%. In the treatment, ascorbic acid, exchange transfusion, and dexamethasone, due to prolonged high values of MetHb, were used. According to Valiev T.T. et al., ifosfamide mustard was the primary cause of this complication [29]. However, it may be worth noting that genetic defects of methemoglobinemia resulting from *CYB5R* deficiency are listed as endemic in some regions of Russia [30]. Unfortunately, the knowledge about genetic *CYB5R* deficiency in our patient is unknown; the patient is of Polish origin.

MetHb concentration and percentage are determined and confirmed using arterial or venous blood gas with co-oximetry, which is a laboratory method of choice to diagnose methemoglobinemia. MetHb has a light absorption peak at 630 nm in a co-oximeter, which is a spectrum absorption analysis device [18]. In the case described by Hadjiliadis et al., MetHb level started from 49.8% with growth in a short time to 56.6% and reduced after 90 mg (1 mg/kg) of MB to 31.2, 24.9, 20.8, 14.6, 10.6, and 1.4% in 1, 2, 3, 5, 7, and 24 h after MB supply [24]. In the reports of Valiev T.T. et al., the examined ALL patients’ levels of MetHb varied from 37.1 to 15% after 24 h of starting the treatment, 31.1 to 1.9, 38.9 to 2.5, 22.9 to 2.4, and 26.5 to 1.7%, respectively [29]. In our patient, MetHb levels were the highest compared to other cases and were 57.1%, half an hour after ifosfamide administration, and after 4 h, the MetHb level fell to 28% following MB injection. Due to the persistently high level of MetHb and the child’s presentation of clinical symptoms, it was necessary to perform an exchange transfusion and administer another MB dose, with the MetHb level being 10.3% after 7 h since the incident started. Furthermore, 18 h after the first symptoms, the MetHb level was 0.7%.

The clinical symptoms of methemoglobinemia are dyspnea, chest pain, headache, dizziness, tachycardia, seizures, central nervous depression, and even death [4,5]. Furthermore, a decrease in saturation may be the initial indicator of hypoxia in severe conditions such as septic shock [31] or pulmonary embolism [32]; nevertheless, if the situation does not improve after administering oxygen, methemoglobinemia should be investigated [2]. Moreover, with our knowledge of the medications used in treatment and their side effects, we may strongly consider methemoglobinemia as a differential diagnosis and then undertake co-oximetry in conjunction with arterial blood gas analysis. In addition, the dark chocolate color of the blood is common for this illness, allowing us to make an accurate diagnosis [18]. The first symptoms in our patient were severe pallor with cyanotic skin, bluish lips, a fine papular rash, sobbing and nervousness with tachycardia reaching HR 190/min, and a decreased saturation of 76–89% with failed attempts at passive oxygen insufflation. Therefore, the child went into respiratory failure and was admitted to the ICU in critical condition with HR 82/min and Sat 70%, requiring intubation and mechanical ventilation. The child’s blood MetHb concentration at the time was 57.1%, and the blood was the color of dark chocolate. After the treatment, the child’s general condition gradually improved. After 2 days in the ICU, the boy was extubated and disconnected from the ventilator. In moderate and stable general condition, he was sent back to the Department of Oncology and Hematology for further treatment.

MB is the first-line therapy for methemoglobinemia [4,5]. The first dose of MB varies from 0.3 mg/kg to 5.5 mg/kg infused over 3 to 5 min, and it can be repeated if symptoms persist within 30 min [5]. Although MB should lower MetHb concentrations within an hour [3], when given in repeated doses, it can worsen methemoglobinemia, and the dose becomes toxic at a total dose of >7 mg/kg [5].

MB is an oxidizing agent that is transformed into leukomethylene blue in the presence of NADPH. Next, the leukomethylene blue converts MetHb to hemoglobin, which enables it to bind to and transport oxygen back through the blood [2,4]. The plasma half-life of MB after intravenous administration is about 5 to 7 h [33].

What is more, if the methemoglobinemia worsens with a poor response to MB therapy, an immediate blood exchange, such as a manual exchange transfusion, should be undertaken [5]. According to the literature, therapeutic whole blood exchange leads to a survival rate of 81.6% in patients refractory to MB [34]. In our case, a possible reason for the failure of MB therapy alone may have been that the concentration of MetHb was too high. Hence, a combination of the effects of MB and an exchange transfusion was necessary to control the child’s condition. In the cases described above and in our patient, treatment with MB and exchange transfusion were effective; the symptoms disappeared, the MetHb level dropped, and patients were discharged home in good condition.

## 4. Conclusions

Methemoglobinemia is an uncommon side effect of ifosfamide administration; however, it should be taken into consideration, especially if the patient exhibits any of the following symptoms: dyspnea with desaturation, cyanosis, chest pain, headache, dizziness, tachycardia, seizures, or nervous central depression with MetHb occurring in co-oximetry. Such a patient requires immediate treatment with a MB injection, and if the therapy fails or the patient’s condition worsens, an exchange transfusion should be performed.

The reasons for methemoglobinemia after ifosfamide administration are not clear. Toxic metabolites of ifosfamide can affect erythrocyte function and potentially alter the degree of iron oxidation. In particular, ifosfamide mustard has a strong affinity for red blood cells, so in the situation of a patient’s genetic susceptibility, rarely such an adverse reaction as methemoglobinemia can occur. Vigilance for this very rare side effect should be widespread when using ifosfamide in treatment protocols.

## Figures and Tables

**Figure 1 ijms-25-03789-f001:**
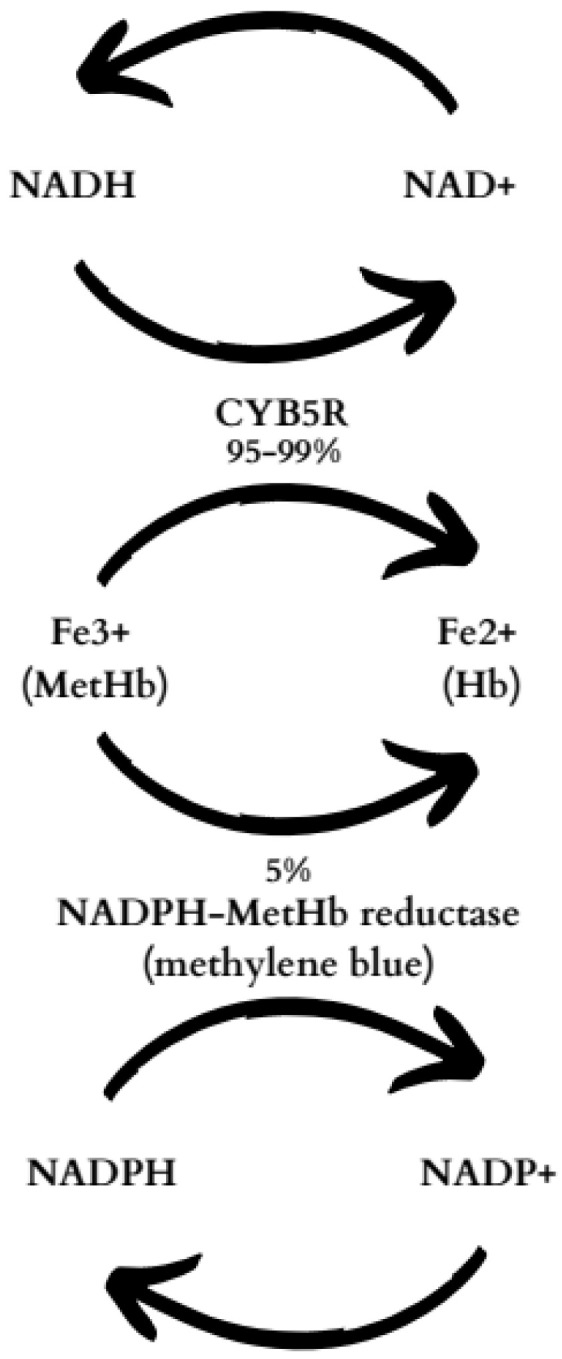
Reduction of methemoglobin—metabolic pathways [1,3,6].

**Figure 2 ijms-25-03789-f002:**
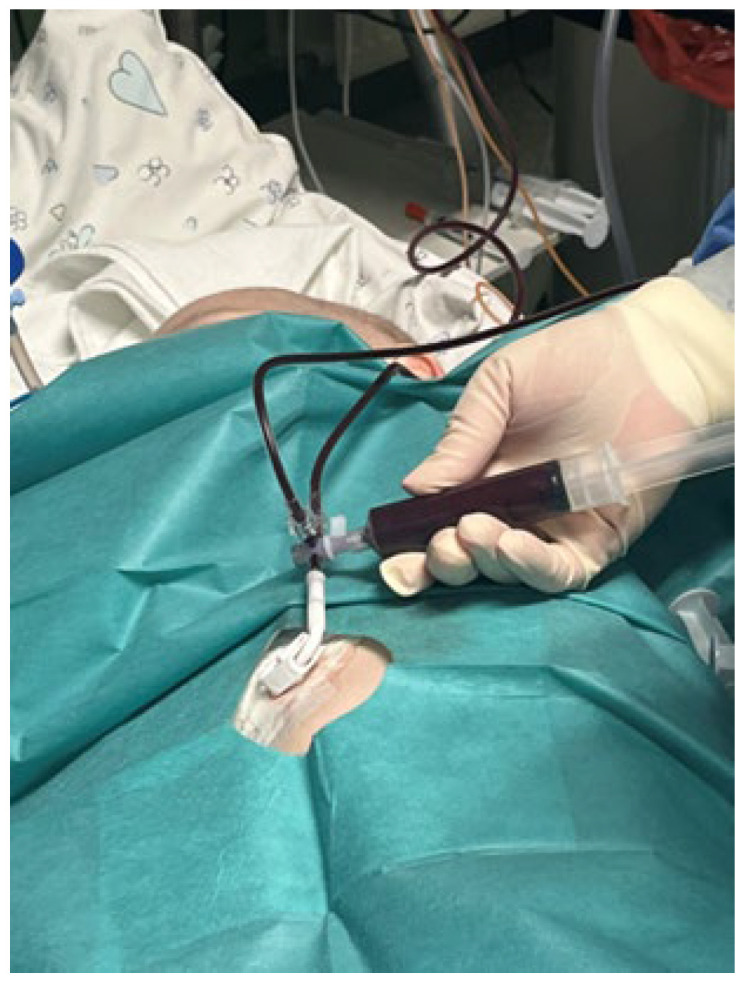
Exchange transfusion. Typical dark brown color of the blood due to methemoglobinemia.

**Figure 3 ijms-25-03789-f003:**
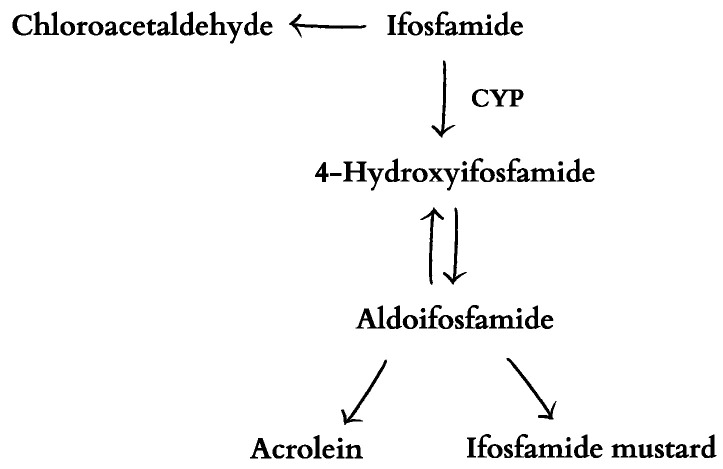
Metabolism of ifosfamide [25,26].

**Table 1 ijms-25-03789-t001:** Blood test results during the methemoglobinemia accident and the following day. MetHb—methemoglobin, O_2_Hb—oxyhemoglobin, HHb—deoxyhemoglobin, COHb—carboxyhemoglobin, THb—total hemoglobin, RBC—red blood cells, HCT—hematocrit, MCV—mean corpuscular volume, PLT—platelets, WBC—white blood cells, VBG—venous blood gases, CRP—C-reactive protein, ALAT—alanine transaminase, ASP—aspartic acid. Values in blue—reduced concentrations compared to the norm; values in red—elevated concentrations compared to the norm. Arrows show MB administration and exchange transfusion during the timeline.

Date:	13 May 2023				14 May 2023
Hour:	0	6	9	23:04	9:12
Methylene blue	↓		↓		
Exchange transfusion		↓			
MetHb [%]	**57.10**	28	10.3	-	0.7
O_2_Hb [%]	42.7	72	87.4	-	98
HHb [%]	0.00	0.00	1.8	-	0.7
COHb [%]	0.2	0.1	0.4	-	0.6
THb [g/dL]	10.38	9.34	9.68	-	11.79
RBC [10^6^/μL]	3.67	-	3.21		3.89
HCT [%]	31.5	-	26.4		32
MCV [fl]	85.8	-	82.2		82.3
PLT [10^3^/μL]	274	-	98		102
WBC [10^3^/μL]	6.38	-	1.17		8.21
Serum glucose [mg/dL]	103	-	-	-	53
VBG:	not available				
blood pH	-	-	7.639	-	7.413
anion gap [mmol/L]	-	-	−2.4	-	−5.3
pCO_2_ [mmHg]	-	-	16.1	-	29.1
pO_2_ [mmHg]	-	-	79	-	14.9
bicarbonates [mmol/L]	-	-	16.9	-	18.1
CRP [mg/L]	<0.6	-	-	-	<0.6
ALAT [U/L]	103	-	-	-	87
ASP [U/L]	-	-	-	-	74
bilirubin [mg/dL]	0.48	-	-	-	0.87
creatinine [mg/dL]	0.43	-	-	-	0.39
urea [mg/dL]	2	-	-	-	-
uric acid [mg/dL]	2	-	-	-	-

## Data Availability

The datasets used during the current study are available from the corresponding author upon reasonable request. The data are not publicly available due to personal data protection, we have eliminated data that would make it possible to identify the patient.

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
