# Peer review of "A Rare Case of Methemoglobinemia after Ifosfamide Infusion in a 3-Year-Old Patient Treated for T-ALL"

_ijms, 2024, doi:10.3390/ijms25073789_

Round 1
Reviewer 1 Report
Comments and Suggestions for Authors
Comments on the case report " A rare case of methemoglobinemia after ifosfamide infusion in a 3-year-old patient treated for T-ALL" ID ijms-2906899
This report provides a complete description of a case of metheglobinemia caused by the administration of isofosfamide in a 3-year-old patient with T-ALL, from the first manifestations to the remission of the symptoms, with emphasis on the treatment and monitoring. Even with the previous knowledge of the association of isofosfamide with some cases of metheglobinemia, I consider valuable this case of particular characteristics as a reference for possible future similar cases. Coincidences and differences in sum are of great value at the time of making therapeutic decisions and even in the configuration of preventive strategies.
Author Response
Dear Reviewer,
Thank you very much for giving us the opportunity to submit a revised draft of our manuscript titled "A rare case of methemoglobinemia after ifosfamide infusion in a 3-year-old patient treated for T-ALL” to the International Journal of Molecular Sciences.
We appreciate the time and effort that you have dedicated to providing your valuable feedback on my manuscript. We are grateful for all the comments on our work. The case of methemoglobinemia caused by the administration of ifosfamide is extremely rare, and we hope that our manuscript will enhance knowledge among doctors using ifosfamide, quicken their reaction, and thus increase the prospects for the patient.
Thank you one more time for your valuable comments.
Kind regards,
Authors

Reviewer 2 Report
Comments and Suggestions for Authors
Certainly informative report. The clinical case, the diagnostic and therapeutic process are well described.
Methemoglobinemia is a rare event, but should be considered in the differential diagnosis in the event of hypoxia associated with cyanosis.I would suggest, for completeness, to also mention the possibility that methemoglobinemia can occur in patients with glucose-6-phosphate dehydrogenase (G6PD) deficiency treated with oxidative stressors. In the case of patients with leukemia, the risk of developing methemoglobinopathy at the onset of the disease for the treatment of tumor lysis syndrome with rasburicase is emblematic
Author Response
Dear Reviewer,
Thank you very much for all the valuable comments that will make important changes to our manuscript entitled "A rare case of methemoglobinemia after infusion of ifosfamide in a 3-year-old patient treated for T-ALL" for the International Journal of Molecular Sciences. We are grateful for your time, thorough analysis and guidance to present the topic in a more detailed and accurate manner. We are grateful for all the comments on our work.
Thank you for confirming that the case we described can help raise awareness among physicians using ifosfamide, and that familiarity with the procedure will contribute to quick response and increased chances for the patient.
We have been able to incorporate changes to reflect most of the suggestions provided by your review.
In answer to comments:
Comment 1: „I would suggest, for completeness, to also mention the possibility that methemoglobinemia can occur in patients with glucose-6-phosphate dehydrogenase (G6PD) deficiency treated with oxidative stressors.”
We posted the information on the potential cause of methemoglobinemia due to G6PD deficiency in the discussion:
“Methemoglobinemia can occur in patients with glucose-6-phosphate dehydrogenase (G6PD) deficiency treated with oxidative stressors. An X-linked gene mutation causes the enzymopathy (G6PD EC 1.1.1.49), which disrupts the function of the sole enzyme in erythrocytes that produces NADPH, the metabolic intermediate essential for the maintenance of a high ratio of reduced to oxidized glutathione that protects erythrocytes from reactive oxygen species-mediated damage.”
Comment 2: „In the case of patients with leukemia, the risk of developing methemoglobinopathy at the onset of the disease for the treatment of tumor lysis syndrome with rasburicase is emblematic”
In Introduction, we added information about hematological conditions where methemoglobinemia can develop due to the treatment, such as raburicase in tumor lysis syndrome:
“When treating tumor lysis syndrome with rasburicase in leukemia patients, the chance of acquiring methemoglobinopathy at the beginning of the illness is significant.”
Thank you one more time for your valuable comments.
Kind regards,
Authors

Reviewer 3 Report
Comments and Suggestions for Authors
The authors present an interesting case and i have the following concerns:
1. What was the precise immunophenotype and the karyotype when the child was diagnosed with T-ALL? Please report.
2. Is the boy still alive? Please report the final outcome.
3. Is it easy from general symptoms to diagnose methemoglobinemia?
What is crucial clinically in order to think of a possible diagnosis of methemoglobinemia and investigate the spectrophotometric method, called co-oximetry?
4. What is the laboratory method of choice for the diagnosis of methemoglobinemia?
5. Are there any other cases of hematological malignancies combined with methemoglobinemia or drug-induuced methemoglobinemia?
Author Response
Dear Reviewer,
Thank you very much for giving us the opportunity to submit a revised draft of our manuscript titled “A rare case of methemoglobinemia after ifosfamide infusion in a 3-year-old patient treated for T-ALL” to the International Journal of Molecular Sciences. We appreciate the time and effort that you have dedicated to providing your valuable feedback on our manuscript. We are grateful for all your comments on our work and undoubtedly it will improve the quality of our case report.
We have been able to incorporate changes to reflect most of the suggestions provided by your review.
In answer to comments:
Comment 1: “1. What was the precise immunophenotype and the karyotype when the child was diagnosed with T-ALL? Please report.”
We added the information about the immunophenotype in the case description part: “Medical diagnosis of T-ALL was based on a bone marrow biopsy with the following cell phenotypes: CD5% - 97%, CD7% - 96%, CD8% - 54%, CD2 - 94% and cytCD3% - 98%. The karyotype of the child has not been performed. Due to poor prednisone response (PPR) and positive minimal residual disease (MRD) in TP1 (time point 1) he was classified to HR group.”
Comment 2: “2. Is the boy still alive? Please report the final outcome.”
We have added information in the Case Report section:
“Currently, the child is during oral maintenance treatment’’
Comment 3a: “3. Is it easy from general symptoms to diagnose methemoglobinemia?”
Comment 3b: “What is crucial clinically in order to think of a possible diagnosis of methemoglobinemia and investigate the spectrophotometric method, called co-oximetry?”
We explained the information about the possibility of the diagnosis of methemoglobinemia in the discussion:
“Furthermore, a decrease in saturation may be the initial indicator of hypoxia in severe conditions such as shock or pulmonary embolism; nevertheless, if the situation does not improve after administering oxygen, methemoglobinemia should be investigated. Moreover, with our knowledge of the medications used in treatment and their side effects, we may strongly consider methemoglobinemia as a differential diagnosis and then undertake co-oximetry in conjunction with arterial blood gas analysis. In addition, the dark chocolate color of the blood is common for this illness, allowing us to make the accurate diagnosis.”
Comment 4: “4. What is the laboratory method of choice for the diagnosis of methemoglobinemia?”
In Discussion, we added information that "co-oximetry is the laboratory method of choice for the diagnosis of methemoglobinemia”.
Comment 5: “5. Are there any other cases of hematological malignancies combined with methemoglobinemia or drug-induuced methemoglobinemia?”
We added information about other medications used in oncology that may cause methemoglobinemia:
“Other medications used during oncology treatment and possibly causing methemoglobinemia are metoclopramide, flutamide, silver nitrate, sulfonamides, cyclophosphamide, and rasburicase. When treating tumor lysis syndrome with rasburicase in leukemia patients, the chance of acquiring methemoglobinopathy at the beginning of the illness is significant. “
Thank you one more time for your valuable comments.
Kind regards,
Authors

Reviewer 4 Report
Comments and Suggestions for Authors
This is a valuable case report in which a child of T-ALL with ifosfamide-induced methemoglobinemia during the chemotherapy was saved by administering methylene blue, and has educational value. Although it has been reported that ifosfamide causes methemoglobinemia, there are not many reports of this. On the contrary, there have been some reports that methylene blue was useful for the treatment of ifosfamide-induced encephalopathy.
In order to accept the case report, I believe that you need to supplement the following two points.
In this case, in a treatment protocol of 5 doses administered every 12 hours of ifosfamide, the onset occurred rapidly during the third dose. There were no signs of any unusual symptoms before the third dose. Please add any thoughts on why the patient had no problems with the previous two doses, but suddenly developed an acute illness during the third dose.
In addition, in this case, treatment was continued with cyclophosphamide as an alternative, but there are clinical reports that similar methemoglobinemia was induced by cyclophosphamide, so why was it continued in this case? I would like you to consider this.
Author Response
Dear Reviewer,
Thank you very much for taking the time to review manuscript titled, “A rare case of methemoglobinemia after ifosfamide infusion in a 3-year-old patient treated for T-ALL” to the International Journal of Molecular Sciences. We appreciate your valuable insight and the effort that you have dedicated to providing your feedback on our manuscript. We are grateful for all the comments on our work.
We have been able to incorporate changes to reflect most of the suggestions provided by your review.
In answer to comments:
Comment 1: “In this case, in a treatment protocol of 5 doses administered every 12 hours of ifosfamide, the onset occurred rapidly during the third dose. There were no signs of any unusual symptoms before the third dose. Please add any thoughts on why the patient had no problems with the previous two doses, but suddenly developed an acute illness during the third dose.”
We added information about the potential cause of the patient's symptoms after the third dose of ifosfamide:
“The mechanism by which the symptoms occurred after administration third dose of ifosfamide in our patient remains unclear. In this case, the likely mechanism may have been the accumulation of the drug dose, as it was administered in short intervals every 12 hours.”
Comment 2: “In addition, in this case, treatment was continued with cyclophosphamide as an alternative, but there are clinical reports that similar methemoglobinemia was induced by cyclophosphamide, so why was it continued in this case? I would like you to consider this.”
We added information on why the patient continued treatment with cyclophosphamide:
“Before the episode of methemoglobinemia patient received 3 doses of 1 g/m2 CPM and 5 doses of 500 mg/m2 CPM without any symptoms of methemoglobinemia and its important alkylating drug used in the treatment of ALL. He continued treatment according to protocol with the use of cyclophosfamide without any symptoms of methemoglobinemia.”
Thank you one more time for your valuable comments.
Kind regards,
Authors

Round 2
Reviewer 3 Report
Comments and Suggestions for Authors
I have no more questions.
Author Response
Dear Reviewer,
We are glad to have answered your questions. Thank you again for all your valuable comments. Thanks to your guidance, our work has been significantly enriched.
Kind regards,
Authors
Reviewer 4 Report
Comments and Suggestions for Authors
Unfortunately, authors have not answered the reviewers' questions in a sufficient manner, so I will ask them again in detail.
The following paper on cyclophosphamide-induced methemoglobinemia should be cited.
Cyclophosphamide-induced methemoglobinemia Bone Marrow Transplant. 2003;32(11):1109-10. doi: 10.1038/sj.bmt.1704278
Additionally, if MetHb was monitored during subsequent cyclophosphamide treatment after the episode of ifosphamide-induced MetHb-nemia, it should be mentioned whether there was a significant increase in MetHb, even though no clinical symptoms were present.
It should be stated whether prophylactic administration of sulfamethoxazole/ trimethoprim was taken during the therapy of ifosphamide, which is universally used for prevention of Pneumocystis jirovecii infection during chemotherapy for ALL, and a well-known drug that could induce MetHb-nemia.

Author Response
Dear Reviewer,
Thank you very much for your valuable comments. We are grateful for your time, thorough analysis, and guidance to present the topic in a more detailed and accurate manner. We have been able to incorporate changes to reflect most of the suggestions provided by your review.
In answer to comments:
Comment 1: Thank you for providing the article: "Cyclophosphamide-induced methemoglobinemia" by Shehadeh et al. It has been tagged with citation number 14.
Comment 2: MetHb levels were not monitored during the administration of cyclophosphamide or cotrimoxazole, however, we were monitoring the saturation levels. We have included the following information in the article:
“Although cyclophosphamide is also recognized as a substance that can cause methemoglobinemia [14], this complication was not observed in our patient. The child had received the drug in previous chemotherapy blocks and was not associated with any complaints suggesting the onset of methemoglobinemia. Continuation of cyclophosphamide treatment was dictated by the patient's normal parameters, and on subsequent administration of the medication, continuously monitored saturation was at a normal level of 95-99%. There was no need to monitor methemoglobin levels. Currently, the child is during oral maintenance treatment. The phenomenon of methemoglobinemia has not appeared again since the discontinuation of ifosfamide administration.”
Comment 3: We have enriched our work with the article “Methemoglobinemia Induced by Trimethoprim-Sulfamethoxazole in a Boy With Acute Lymphoblastic Leukemia” by Lian et al. (13) Thank you for pointing out to us the need to single out this drug as potentially causing methemoglobinemia. However, our patient received this drug in between chemotherapy blocks and it was not associated with the appearance of methemoglobinemia symptoms. During the event described, the patient was not taking this medication. We described it as follows:
“One drug that is well-known to cause methemoglobinemia is trimethoprim-sulfamethoxazole used to prevent Pneumocystis jirovecii in patients treated for ALL [13]. However, our patient received prophylactic treatment in between blocks of chemotherapy and no complaints were observed from any administration of this medication.”
Once again, thank you for all your guidance and valuable comments, which will certainly increase the value of our work.
Kind regards,
Authors